# VideoTitans: Scalable Video Prediction with Integrated Short- and Long-term Memory

**Young-Jae Park**
Department of AI Convergence
GIST
youngjae.park@gm.gist.ac.kr

**Minseok Seo**
School of Electrical Engineering
KAIST
minseok.seo@kaist.ac.kr

**Hae-Gon Jeon**[*]
Department of Artificial Intelligence
Yonsei University
earboll@yonsei.ac.kr

## Abstract

Accurate video forecasting enables autonomous vehicles to anticipate hazards, robotics and surveillance systems to predict human intent, and environmental models to issue timely warnings for extreme weather events. However, existing methods remain limited: transformers rely on global attention with quadratic complexity, making them impractical for high-resolution, long-horizon video prediction, while convolutional and recurrent networks suffer from short-range receptive fields and vanishing gradients, losing key information over extended sequences. To overcome these challenges, we introduce *VideoTitans*, the first architecture to adapt the gradient-driven *Titans* memory—originally designed for language modelling to video prediction. VideoTitans integrates three core ideas: (i) a sliding-window attention core that scales linearly with sequence length and spatial resolution, (ii) an episodic memory that dynamically retains only informative tokens based on a gradient-based *surprise* signal, and (iii) a small set of persistent tokens encoding task-specific priors that stabilize training and enhance generalization. Extensive experiments on Moving-MNIST, Human3.6M, TrafficBJ and WeatherBench benchmarks show that VideoTitans consistently reduces computation (FLOPs) and achieves competitive visual fidelity compared to state-of-the-art recurrent, convolutional, and efficient-transformer methods. Comprehensive ablations confirm that each proposed component contributes significantly.

## 1 Introduction

Accurate video forecasting enables proactive decision-making in critical real-world systems such as autonomous driving [22, 24, 73], city-scale surveillance [15, 40, 41], robotics [29, 25, 52], and weather forecasting [18, 42, 43]. Predicting future video frames demands a delicate balance: the model must precisely capture rapid, subtle changes between frames [6], yet retain memory of important events and scene dynamics over extended time horizons. Traditional approaches rely heavily on convolutional [28, 61, 54, 69] or recurrent architectures [66, 71], which handle local dynamics effectively but face challenges due to limited receptive fields [7, 47] and vanishing gradients [64, 33], severely restricting their performance on long sequences.

---

[*]Corresponding author

39th Conference on Neural Information Processing Systems (NeurIPS 2025).

Recent transformer-based architectures address these limitations by employing global self-attention [1, 32, 9, 21] to capture long-range dependencies. However, this comes at the prohibitive cost of quadratic computational complexity [76, 3], making them infeasible for realistic, high-resolution, long-sequence applications. Attempts to circumvent this computational bottleneck—such as hierarchical window attention [38], low-rank approximation [34], or external memory [12]—introduce rigid architectural constraints and task-specific heuristics, limiting their generalizability and flexibility across diverse forecasting scenarios.

An alternative approach emerges from recent advances in natural-language processing. Titans [2], a gradient-driven episodic memory module, selectively commits information to memory only when its loss gradient signals substantial "surprise"—a mechanism motivated by the way humans tend to remember unexpected or novel events [35]. This memory mechanism naturally aligns with video prediction tasks, where redundant frame sequences dominate, punctuated by critical rare events such as sudden object movements or abrupt camera motions. However, adapting Titans directly to video forecasting is non-trivial: handling high-resolution frames substantially inflates memory complexity, standard transformer attention remains a computational bottleneck, and visual forecasting benefits significantly from learned, static priors which episodic memory alone cannot provide.

In this paper, we introduce *VideoTitans*, the first architecture to successfully adapt the Titans gradient-driven memory to the dense video forecasting domain. VideoTitans uniquely integrates three core components into a unified, computationally efficient framework: (i) a lightweight sliding-window attention core whose complexity grows linearly with sequence length and spatial resolution, (ii) a gradient-based episodic memory that selectively encodes surprising patch tokens, and (iii) a small set of persistent tokens that inject input-agnostic, reusable priors into the prediction pipeline. The interplay of these modules is seamlessly coordinated by a single gating mechanism, ensuring the predictor remains end-to-end differentiable without reliance on manually tuned heuristics.

We conduct extensive evaluations across diverse and challenging benchmarks—Moving-MNIST [53], Human3.6M [26], TrafficBJ [74], and WeatherBench [45]—demonstrating that VideoTitans consistently reduces computational load while delivering superior visual fidelity in long-range forecasts compared to state-of-the-art recurrent, convolutional, and efficient-transformer methods. Our comprehensive ablation studies further verify that each proposed component is critical to achieving this performance. To facilitate further research and ensure full reproducibility, we will publicly release our source code, trained checkpoints, and demonstration videos.

**Contributions**

- We demonstrate that gradient-driven Titans memory can be applied to long, high-resolution video sequences without incurring quadratic computational growth, providing the first cross-domain evidence of its effectiveness beyond language.
- We present a unified memory–attention architecture that balances efficiency and temporal coverage by combining sliding-window attention, episodic memory and persistent priors behind a single gating mechanism.
- Extensive experiments on Moving-MNIST, Human3.6M, TrafficBJ and WeatherBench show consistent reductions in computation and improvements in long-range visual fidelity over state-of-the-art recurrent, convolutional, and efficient-transformer baselines, while ablation studies confirm that every component of VideoTitans is indispensable.

## 2  Related Works

### 2.1  Memory in RNNs and Transformers

Recurrent neural networks (RNNs) [70, 36, 55] and their variants, such as Long Short-Term Memory (LSTM) [23, 48, 77] and Gated Recurrent Units (GRU) [10, 11, 13], have been widely used for modeling sequential dependencies in video prediction [37, 39, 16, 71]. These architectures utilize internal memory to retain past information, enabling them to capture long-range dependencies [56, 31]. However, they suffer from vanishing gradients and sequential processing constraints, limiting their scalability to long video sequences [44]. While various enhancements have been proposed to improve memory retention, RNN-based approaches remain computationally inefficient for high-dimensional spatio-temporal modeling [27, 5].

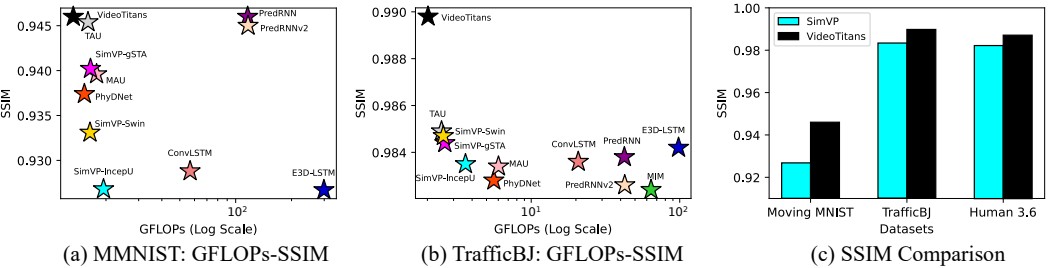

(a) MMNIST: GFLOPs-SSIM    (b) TrafficBJ: GFLOPs-SSIM    (c) SSIM Comparison

Figure 1: Performance and efficiency comparisons among video prediction models. Compared to other video prediction models on benchmark datasets, VideoTitans achieves lower FLOPs(G) while delivering superior or comparable performance.

Transformer-based architectures have gained prominence for their ability to model long-term dependencies through self-attention mechanisms, enabling parallelized sequence processing [62, 60, 20]. However, standard attention scales quadratically with sequence length, making it impractical for long video sequences. To address the issue, memory-augmented transformers incorporate external memory to store and retrieve key representations, reducing computational overhead while preserving global contextual information [72, 4, 30]. Despite these improvements, challenges remain in retrieval efficiency and adaptability to dynamic dependencies [59, 63]. Our work builds upon these advancements by introducing a neural long-term memory module that selectively retains critical past information, improving both efficiency and predictive robustness for video forecasting.

## 2.2 Video Prediction

Video prediction involves forecasting future frames based on past observed frames by modeling intricate spatio-temporal dependencies. ConvLSTM [51] introduced convolutional recurrent units to jointly capture spatial and temporal contexts but struggled with long-term stability. PredRNN [64] and its variants [65, 68] significantly improved temporal modeling by incorporating additional spatio-temporal memory units but came with considerable computational overhead. E3D-LSTM [66] further enhanced performance by integrating 3D convolutions, yet remained computationally demanding. PhyDNet [19] leveraged physical constraints to better represent motion dynamics but was limited in modeling highly complex scenarios.

SimVP [17] significantly simplified the prediction model by employing spatial-temporal separable architectures, balancing performance with computational efficiency. Building upon this, SimVP-meta [58] extended SimVP by integrating recurrent, convolutional, and transformer-based architectures into a unified meta-model framework, greatly advancing the field of video prediction. Following this work, the autoregressive-based [50] model has further enriched the literature with promising directions.

Inspired by these developments, our paper introduces, for the first time, a novel Titans-based [2] architecture—*VideoTitans*—specifically designed to enhance both long-term and short-term memory capabilities, thereby addressing critical challenges in video prediction tasks.

## 3 Preliminaries

**Memory and Sequence Modeling.**    Sequential modeling tasks, such as video prediction, typically involve handling long-term temporal dependencies. Recurrent neural networks (RNNs) encode these dependencies into a compressed hidden state but often lose essential information over longer sequences. On the other hand, Transformers explicitly model dependencies using attention, but this comes with quadratic complexity, limiting their applicability to very long sequences such as videos.

**Neural Memory and Adaptive Forgetting.**    Recent architectures introduce explicit neural memory modules that dynamically store historical context beyond the immediate attention window. These modules update memory state $\mathbf{M}_t$ through recursive formulations such as:

$$\mathbf{M}_t = f(\mathbf{M}_{t-1}, \mathbf{x}_t), \tag{1}$$

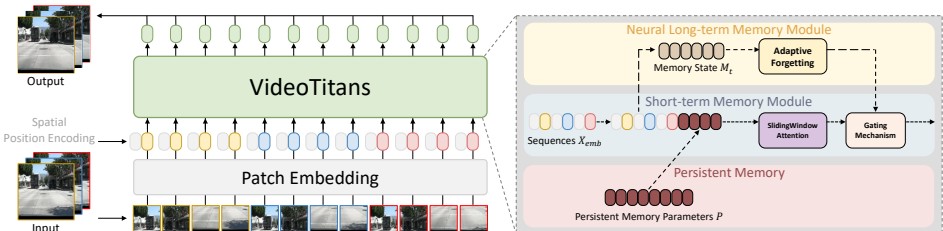

Figure 2: Overview of the VideoTitans framework, which integrates neural long-term memory, sliding window attention, and persistent memory to enhance video prediction. The model dynamically adapts to both short-term and long-term dependencies through a gradient-based surprise mechanism. The final prediction is obtained by combining short-term attention and memory states via a gating mechanism.

where $f$ represents the memory update function that selectively encodes relevant information while employing adaptive forgetting mechanisms to discard redundant details. Such adaptive mechanisms are crucial to managing long-term dependencies without memory overflow.

**Titans Framework.** Titans proposes a neural memory module specifically designed to dynamically learn, memorize, and retrieve crucial information. Titans define a gradient-based surprise mechanism to determine the importance of events:

$$\mathbf{M}_t = \mathbf{M}_{t-1} + \mathbf{S}_t, \quad \mathbf{S}_t = \eta_t \mathbf{S}_{t-1} - \theta_t \nabla \ell(\mathbf{M}_{t-1}; \mathbf{x}_t), \tag{2}$$

where $\mathbf{S}_t$ captures both historical (past) and immediate (momentary) surprise. This allows Titans to dynamically adapt and selectively encode important events, balancing short-term attention and long-term memorization effectively.

Inspired by this approach, we introduce *VideoTitans*, adapting the Titans memory framework to video forecasting tasks, allowing efficient modeling of both local and global temporal dependencies inherent in video data.

## 4 Methodology

### 4.1 Problem Definition.

Given an input video sequence $\mathbf{X} \in \mathbb{R}^{B \times T \times C \times H \times W}$ consisting of $T$ observed frames, the goal of video forecasting is to accurately predict subsequent future video frames $\mathbf{Y} \in \mathbb{R}^{B \times \hat{T} \times C \times H \times W}$. Here, $B$ represents the batch size, $T$ denotes the number of observed frames, $\hat{T}$ is the number of future frames to predict, $C$ corresponds to the number of channels, and $H, W$ indicate the spatial dimensions (height and width) of each frame. Formally, the task can be defined as learning a function:

$$\hat{\mathbf{Y}} = \mathcal{F}(\mathbf{X}; \theta), \tag{3}$$

where the model $\mathcal{F}$, parameterized by $\theta$, aims to learn complex spatio-temporal dependencies from past video frames and leverage them to generate precise, high-fidelity predictions for future frames. The challenge is that video data inherently contains both short-term dynamics (local correlations between consecutive frames) and long-term dependencies (slowly evolving or periodic patterns across multiple frames), making the accurate modeling of both short-term and long-term temporal relationships critical for reliable predictions.

### 4.2 VideoTitans

**Input Embedding and Positional Encoding.** The input video sequence $(B, T, C, H, W)$ is reshaped into $(B \times T, C, H, W)$. Each frame is embedded into spatial patches with positional encoding (PE):

$$(B \times T, C, H, W) \to (B \times T, \text{embed\_dim}, H/16, W/16).$$

We then permute the embeddings into a form suitable for temporal modeling: $(B \times T, \text{embed\_dim}, H/16, W/16) \to (B, H/16 \times W/16, T \times \text{embed\_dim})$.

| | MovingMNIST | | | | TrafficBJ | | | | Human3.6 | | | |
|---|---|---|---|---|---|---|---|---|---|---|---|---|
| Method | MSE(↓) | MAE(↓) | SSIM(↑) | FLOPs(G) | MSE(↓) | MAE(↓) | SSIM(↑) | FLOPs(G) | MSE(↓) | MAE(↓) | SSIM(↑) | FLOPs(G) |
| ConvLSTM [51] | 29.7990 | 90.6396 | 0.9288 | 56.8 | 0.3358 | 15.3175 | 0.9836 | 20.74 | 125.5210 | 1566.7080 | 0.9813 | 347.0 |
| E3D-LSTM [66] | 38.5383 | 83.6387 | 0.9267 | 298.9 | 0.3427 | 14.9824 | 0.9842 | 98.19 | 143.3489 | 1442.4492 | 0.9803 | 542.0 |
| MIM [67] | 22.5508 | 69.9673 | **0.9488** | 179.2 | 0.3130 | 14.9387 | 0.9824 | 64.10 | 111.8432 | 1463.4142 | 0.9830 | 1051.0 |
| PhyDNet [19] | 28.1955 | 78.6397 | 0.9374 | 15.3 | 0.3622 | 15.5315 | 0.9828 | 5.60 | 125.7428 | 1614.7234 | 0.9804 | **19.1** |
| PredRNN [64] | 23.9667 | 72.8222 | 0.9460 | 116.0 | 0.3194 | 15.3077 | 0.9838 | 42.40 | 113.1855 | 1458.3422 | 0.9831 | 704.0 |
| PredRNNv2 [68] | 24.1136 | 73.7252 | 0.9450 | 116.6 | 0.3834 | 15.5528 | 0.9826 | 42.63 | 114.8799 | 1484.8729 | 0.9827 | 708.0 |
| MAU [8] | 26.8564 | 78.2186 | 0.9396 | 17.8 | 0.3268 | 15.2582 | 0.9834 | 6.02 | 127.3176 | 1577.0112 | 0.9812 | 105.0 |
| TAU [57] | 24.6029 | 71.9298 | 0.9454 | 16.0 | 0.3108 | 14.9341 | 0.9849 | 2.49 | 113.3487 | **1390.6997** | 0.9839 | 182.0 |
| SimVP-IncepU [17] | 32.1478 | 89.0498 | 0.9268 | 19.4 | 0.3282 | 15.4554 | 0.9835 | 3.61 | 115.8376 | 1511.4755 | 0.9822 | 197.0 |
| SimVP-gSTA [58] | 26.6926 | 77.1883 | 0.9402 | 16.5 | 0.3247 | 15.0290 | 0.9844 | 2.62 | **108.0713** | 1444.5731 | 0.9833 | 74.6 |
| SimVP-Swin [58] | 29.6991 | 84.0507 | 0.9331 | 16.4 | 0.3127 | 15.0689 | 0.9847 | 2.56 | 133.2034 | 1599.7281 | 0.9799 | 188.0 |
| SimVP-Uniformer [58] | 30.3827 | 85.8719 | 0.9308 | 16.5 | 0.3268 | 15.1653 | 0.9844 | 2.71 | 116.3079 | 1497.6663 | 0.9824 | 211.0 |
| SimVP-ViT [58] | 35.1473 | 95.8649 | 0.9140 | 16.9 | 0.3171 | 15.1532 | 0.9841 | 2.80 | 136.3321 | 1603.5026 | 0.9796 | 239.0 |
| SimVP-Poolformer [58] | 31.7882 | 88.4830 | 0.9271 | 14.1 | 0.3273 | 15.3947 | 0.9840 | 2.06 | 118.4458 | 1484.1716 | 0.9827 | 156.0 |
| VideoTitans | **21.3265** | **65.2124** | 0.9463 | 13.33 | **0.3099** | 14.8220 | **0.9898** | **1.90** | 109.4821 | 1401.3456 | **0.9871** | 128.52 |

Table 1: A performance comparison of VideoTitans with other approaches is conducted on three standard benchmark datasets for future frame prediction. VideoTitans consistently achieves competitive results across Moving MNIST, TrafficBJ, and Human 3.6, which differ in characteristics.

**Neural Long-term Memory Module.** We introduce an adaptive neural long-term memory module based on a gradient-based surprise mechanism. The memory state update rule at time $t$ is:

$$\mathbf{M}_t = (1 - \alpha_t)\mathbf{M}_{t-1} + \mathbf{S}_t, \tag{4}$$

where the surprise score $\mathbf{S}_t$ is computed by:

$$\mathbf{S}_t = \eta_t \mathbf{S}_{t-1} - \theta_t \nabla \ell(\mathbf{M}_{t-1}; \mathbf{x}_t). \tag{5}$$

Here, parameters $\alpha_t$, $\eta_t$, and $\theta_t$ control adaptive forgetting, surprise decay, and momentary surprise integration, respectively, enabling the selective memorization of crucial historical information.

**Sliding Window Attention.** To precisely model short-term temporal dependencies, sliding window attention is applied to embedded input sequences:

$$\mathbf{Y}_S = \text{SlidingWindowAttention}(\mathbf{X}_{emb}), \tag{6}$$

where $\mathbf{X}_{emb}$ denotes the reshaped spatial embeddings.

**Persistent Memory.** To encode task-specific and context-independent information, we incorporate persistent memory parameters $\mathbf{P}$. These parameters are concatenated to the embedded input as follows:

$$\mathbf{X}_{new} = [\mathbf{p}_1, \mathbf{p}_2, \ldots, \mathbf{p}_{N_p}] \| \| \mathbf{X}_{emb}. \tag{7}$$

**Decoding and Frame Reconstruction.** The final prediction is obtained by decoding the combined representations of short-term attention and neural long-term memory through a gating mechanism:

$$\hat{\mathbf{Y}} = \text{Decoder}\left(\sigma(\mathbf{Y}_S \otimes \mathbf{M}_t)\right). \tag{8}$$

The decoded output $\hat{\mathbf{Y}}$ is reshaped back to the original video dimensions $(B, \hat{T}, C, H, W)$.

## 5 Experiments

In this section, we present extensive evaluations of our proposed VideoTitans architecture on widely adopted benchmarks for future frame prediction. Additionally, we analyze the effectiveness of each component of VideoTitans through comprehensive ablation studies.

| Dataset | $N_{\text{train}}$ | $N_{\text{test}}$ | $(C, H, W)$ | $T$ | $T'$ |
|---|---|---|---|---|---|
| Moving MNIST [53] | 10,000 | 10,000 | (1, 64, 64) | 10 | 10 |
| TrafficBJ [74] | 20,461 | 500 | (2, 32, 32) | 4 | 4 |
| Human3.6 [26] | 73,404 | 8,582 | (3, 128, 128) | 4 | 4 |
| Weatherbench [45] | 2,167 | 706 | (1/2, 32, 64) | 12 | 12 |

Table 2: Summary of datasets used. $N_{\text{train}}/N_{\text{test}}$ are sample counts. $(C, H, W)$: input shape. $T/T'$: input/predicted frames.

**Datasets** We evaluate VideoTitans on four widely-used datasets for future frame prediction, summarized in Table 2: Moving MNIST (MMNIST) [53], TrafficBJ [74], Human 3.6 [26], and WeatherBench [45]. Moving MNIST consists of synthetically generated video sequences depicting

two digits moving randomly within a constrained grid, making it ideal for evaluating models on simple nonlinear temporal dynamics. TrafficBJ contains real-world traffic flow data collected in Beijing, capturing complex urban spatio-temporal dynamics. Human 3.6M comprises high-resolution motion capture data of human activities, challenging the model to accurately capture subtle and intricate human motions. WeatherBench includes global meteorological variables such as temperature (`t2m`), wind fields (`uv10`), and total cloud cover (`tcc`), testing the model's capacity to handle complex global-scale spatio-temporal interactions.

**Evaluation Metric**   We use widely adopted metrics to assess prediction quality and evaluate four different datasets. Specifically, MMNIST, TrafficBJ, and Human3.6 are evaluated using the Mean Square Error (MSE), Mean Absolute Error (MAE), and Structural Similarity Index Measure (SSIM). The weatherbench is evaluated with MSE, MAE, and Root Mean Square Error (RMSE).

**Implementation Details**   Following [58], we optimize VideoTitans using the Adam optimizer and train with the Mean Squared Error (MSE) loss. We set the batch size to 8 for all experiments. The learning rate is adaptively adjusted using the ReduceLROnPlateau scheduler with patience of 10 epochs. Initial learning rates are selected from the set $\{10^{-2}, 5 \times 10^{-3}, 10^{-3}, 5 \times 10^{-4}, 10^{-4}\}$, and the best-performing value is used for each dataset. The total number of training epochs varies depending on the dataset complexity and size. All experiments are implemented using PyTorch and conducted on 8 NVIDIA A100 GPUs. More details can be found in the supplementary material and the code.

## 5.1   Quantitative results

**Moving MNIST**   The Moving MNIST dataset is characterized by simple yet highly nonlinear dynamics involving continuous movements and interactions of digit shapes. Our VideoTitans effectively captures these nonlinear temporal dynamics, demonstrating state-of-the-art performance in long-term forecasting by adaptively preventing error accumulation. This indicates the robustness of our model in handling synthetic nonlinear trajectories. Detailed quantitative results are provided in the left column of Table 1.

**TrafficBJ**   The TrafficBJ dataset contains real-world urban traffic sequences exhibiting complex spatio-temporal patterns and periodic fluctuations. VideoTitans successfully captures both short-term local variations and long-term global trends, achieving superior forecasting accuracy compared to transformer-based and recurrent models. This highlights the applicability of VideoTitans to dynamic urban traffic scenarios. Comprehensive performance comparisons are provided in the middle column of Table 1.

**Human 3.6M**   The Human 3.6M dataset includes sequences of articulated human motions captured under controlled conditions, demanding precise modeling of intricate spatio-temporal interactions. VideoTitans demonstrates robust performance by effectively modeling subtle short-term movements while maintaining long-term coherence in human motion sequences. These results underscore its strength in detailed human motion prediction. Full comparisons are available in the right column of Table 1. We further include results on the Caltech-Pedestrian [14] in the Supplementary Material.

**WeatherBench**   The WeatherBench dataset involves forecasting long-range meteorological variables such as temperature (`t2m`), wind fields (`uv10`), and total cloud cover (`tcc`), characterized by complex spatio-temporal dependencies and nonlinear interactions on a global scale. VideoTitans demonstrates robust performance on these variables, as detailed in Table 3, achieving competitive performance compared to previous methods with lower FLOPs. This performance suggests that VideoTitans effectively

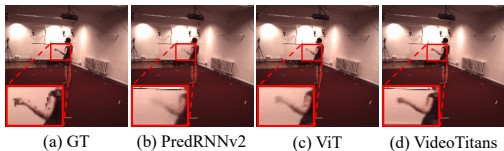

| (a) GT | (b) PredRNNv2 | (c) ViT | (d) VideoTitans |

Figure 3: Qualitative comparison of predicted future frames on the Human3.6 dataset. Our model makes more accurate predictions, particularly noticeable in the person's arm.

captures large-scale dependencies inherent in meteorological data, making the model potentially suitable for weather forecasting tasks. Nonetheless, since VideoTitans does not explicitly model

| Method | t2m MSE(↓) | MAE(↓) | RMSE(↓) | FLOPs(G) | uv10 MSE(↓) | MAE(↓) | RMSE(↓) | FLOPs(G) | tcc MSE(↓) | MAE(↓) | RMSE(↓) | FLOPs(G) |
|---|---|---|---|---|---|---|---|---|---|---|---|---|
| ConvLSTM | 1.9866 | 0.8558 | 1.4095 | 136 | 2.4170 | 1.0180 | 1.5547 | 136 | 0.0722 | 0.1746 | 0.2688 | 136 |
| E3D-LSTM | 1.5921 | 0.8059 | 1.2618 | 169 | 2.4111 | 1.0341 | 1.5528 | 171 | 0.0573 | 0.1529 | 0.2394 | 169 |
| MIM | 2.3940 | 0.9837 | 1.5473 | 109 | 3.3708 | 1.2363 | 1.8360 | 109 | 0.0798 | 0.1836 | 0.2825 | 109 |
| PhyDNet | 290.9133 | 8.8492 | 17.0496 | 36.8 | 16.7869 | 2.9188 | 4.0972 | 36.8 | 0.0997 | 0.2261 | 0.3157 | 36.8 |
| PredRNN | 1.7250 | 0.7987 | 1.3134 | 278 | 2.6378 | 1.0804 | 1.6241 | 279 | 0.0789 | 0.1803 | 0.2810 | 278 |
| PredRNN++ | 1.4575 | 0.7676 | 1.2073 | 413 | 2.5476 | 1.0548 | 1.5961 | 414 | 0.0797 | 0.1954 | 0.2824 | 413 |
| PredRNNv2 | 1.7826 | 0.8074 | 1.3351 | 279 | 2.8591 | 1.1303 | 1.6909 | 280 | 0.0828 | 0.1874 | 0.2878 | 279 |
| MAU | 1.2413 | 0.6977 | 1.1141 | 39.6 | 2.1530 | 0.9594 | 1.4673 | 39.6 | 0.0707 | 0.1715 | 0.2660 | 39.6 |
| TAU | 1.3611 | 0.7056 | 1.1667 | 6.70 | 1.7051 | 0.8509 | 1.3058 | 6.70 | 0.0661 | 0.1653 | 0.2570 | 6.70 |
| SimVP-IncepU | 1.7897 | 0.8015 | 1.3378 | 8.03 | 1.9993 | 0.9510 | 1.4140 | 8.04 | 0.0754 | 0.1760 | 0.2747 | 8.03 |
| SimVP-gSTA | **1.1523** | **0.6524** | **1.0735** | 7.01 | 1.7272 | 0.8812 | 1.3142 | 7.02 | **0.0469** | **0.1474** | **0.2166** | 7.01 |
| SimVP-Swin | 1.2235 | 0.6665 | 1.1061 | 6.88 | 1.5709 | 0.8168 | 1.2533 | 6.89 | 0.0589 | 0.1567 | 0.2426 | 6.88 |
| SimVP-Uniformer | 1.1948 | 0.6697 | 1.0930 | 7.45 | 1.4781 | 0.8059 | 1.2158 | 7.46 | 0.0561 | 0.1553 | 0.2368 | 7.45 |
| SimVP-ViT | 1.2954 | 0.6873 | 1.1382 | 7.99 | 1.6893 | 0.8512 | 1.2997 | 8.0 | 0.0615 | 0.1596 | 0.2480 | 7.99 |
| SimVP-Poolformer | 1.2525 | 0.6711 | 1.1191 | 5.61 | 1.6678 | 0.8427 | 1.2914 | 5.62 | 0.0562 | 0.1530 | 0.2371 | 5.61 |
| VideoTitans | 1.1852 | 0.6636 | 1.1158 | **4.92** | **1.4056** | **0.7984** | **1.1850** | **4.92** | 0.0554 | 0.1522 | 0.2353 | **4.92** |

Table 3: Performance comparison of VideoTitans and state-of-the-art methods on the WeatherBench dataset for predicting temperature (t2m), wind velocity (uv10), and total cloud cover (tcc). VideoTitans demonstrates competitive predictive accuracy across all variables, effectively capturing complex global spatio-temporal patterns inherent in weather forecasting tasks.

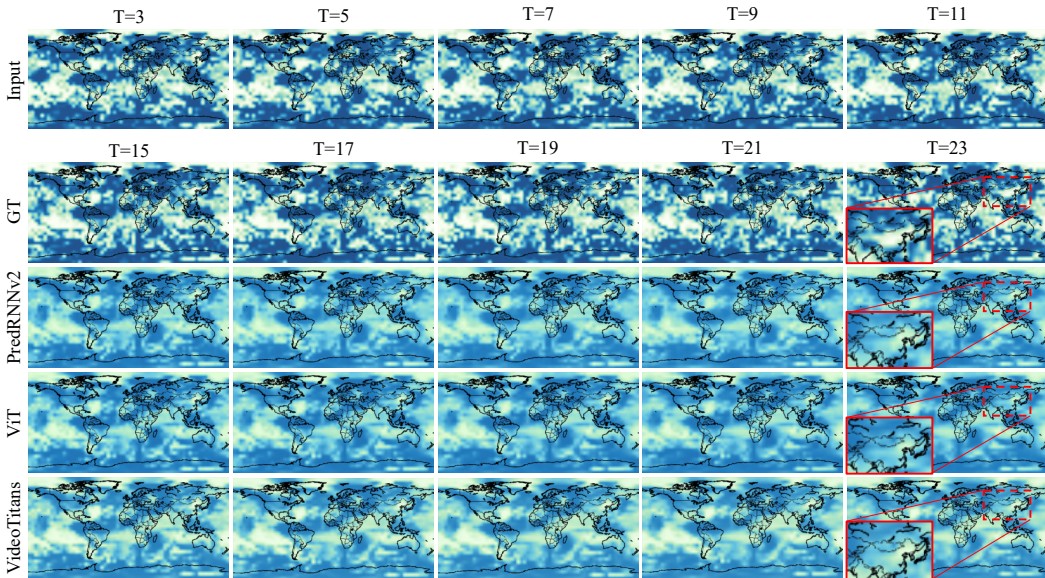

Figure 4: Qualitative comparison of predicted total cloud cover (tcc) frames on the WeatherBench dataset. Predictions from VideoTitans are compared against PredRNNv2 (recurrent-based) and ViT (transformer-based). Red boxes highlight regions where VideoTitans better preserves cloud patterns compared to other methods, indicating its capability to effectively model global spatio-temporal interactions in meteorological data.

the continuous fine-grained dynamics common in atmospheric processes, additional architectural improvements could further enhance its predictive capabilities for subtle climate interactions.

## 5.2 Qualitative results

**Human3.6** Figure 3 presents qualitative comparisons between VideoTitans and baseline methods on the Human 3.6 dataset. The superiority of VideoTitans is clearly evident, as it generates sharper and more accurate predictions for subtle and articulated human movements compared to recurrent (PredRNNv2) and transformer-based (ViT) methods. This highlights VideoTitans' capability to effectively capture detailed motion dynamics and maintain prediction quality.

**WeatherBench** We evaluate VideoTitans on the WeatherBench dataset using the standard protocol from OpenSTL, predicting weather variables at 1-hour intervals up to 12 hours into the future. The qualitative results in Figure 4 visualize the predicted total cloud cover (tcc) at the reduced resolution following the OpenSTL standard experimental setup. Since our primary objective is general video

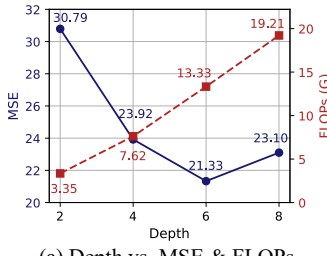
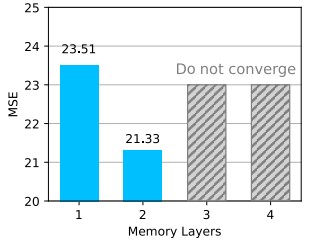
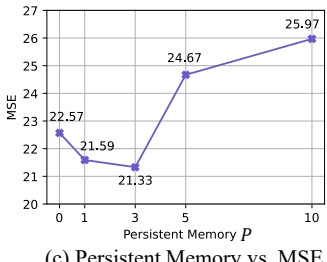

| (a) Depth vs. MSE & FLOPs | (b) Memory Layers vs. MSE | (c) Persistent Memory vs. MSE |

Figure 5: Ablation analysis of VideoTitans on the Moving MNIST dataset. (a) illustrates how model memory depth affects both FLOPs and MSE, showing that moderate depth offers the best trade-off between efficiency and accuracy. (b) presents the effect of the number of neural memory layers, where performance peaks at two layers, while deeper configurations encounter training instability. (c) shows the impact of persistent memory parameters $\mathbf{P}$, indicating that selecting the optimal value is key to achieving the best performance.

prediction, we opt for this protocol; however, future work will extend our evaluation to the higher-resolution WeatherBench2 [46] dataset at its full spatial resolution (0.25-degree). This will allow us to better examine VideoTitans' capability in capturing finer-grained meteorological dynamics. Additional qualitative results across datasets are provided in the Supplementary Material.

## 5.3 Ablation Study

**Effect of Memory Depth** We analyze the effect of varying the depth (number of attention blocks) of the VideoTitans on the Moving MNIST dataset in figure 5 (a). Increasing the depth initially improves prediction accuracy, reaching the lowest MSE at a depth of 6. However, beyond this depth, we observe diminishing returns, as the MSE slightly increases at depth 8, suggesting a trade-off between computational complexity and prediction accuracy. These results indicate that a depth of 6 achieves the optimal balance between model complexity (measured in FLOPs) and forecasting performance.

| Hyperparameter | Value |
|---|---|
| Neural Memory Depth | 2 |
| Neural Demory Dim | 512 |
| Head | 4 |
| Momentum Order | 1 |
| Max Gradient Norm | 1.0 |
| Persistent Mem Tokens | 4 |
| Chunk Size | 256 |
| Segment Len | 256 |
| Long-term Mem Tokens | 16 |

Table 4: Detailed hyperparameter configuration used for VideoTitans training.

**Design of Neural memory** We investigate the effect of neural memory depth on the prediction performance of VideoTitans by varying the number of layers within the memory module from 1 to 4 in figure 5 (b). The model achieves the best performance (MSE: 21.3265) with a memory depth of 2. Despite extensive experimentation—including careful tuning of hyperparameters, gradient norm clipping, learning rate adjustments, and various initialization strategies—deeper memory modules (3 and 4 layers) consistently faced severe training instabilities and failed to converge. This highlights a critical trade-off between memory depth and training stability, indicating that very deep neural memory structures may require extensive and precise hyperparameter tuning or architectural modifications to ensure convergence and maintain stability.

**Influence of Persistent Memory Parameter ($\mathbf{P}$)** Figure 5 (c) examines how varying the Persistent Memory Parameter ($\mathbf{P}$) influences the prediction performance. The optimal performance (MSE: 21.3265) is achieved at $\mathbf{P} = 3$. Smaller values ($\mathbf{P} = 0$ or 1) and larger values ($\mathbf{P} = 5$ or 10) degrade performance, suggesting that a moderate value of $\mathbf{P}$ balances model complexity and memory capacity for the best forecasting outcomes.

**Effect of Persistent Memory** Table 5 evaluates the impact of including Persistent Memory in VideoTitans on the Moving MNIST dataset. The presence of Persistent Memory significantly reduces MSE from 23.5125 to 21.3265, indicating that Persistent Memory effectively helps the model retain critical temporal context, leading to improved prediction accuracy.

| Persistent Memory | MSE |
|---|---|
| With | **21.3** |
| Without | 23.5 |

Table 5: Persistent Memory ablation on Moving MNIST.

| Attention Type | MSE |
|---|---|
| Sliding Window | **21.3** |
| Global Attention | 24.4 |
| No Attention | 30.7 |

Table 6: Attention mechanism ablation on Moving MNIST.

| Method | MSE |
|---|---|
| Memory as a Context | 66.5201 |
| Memory as a Gate | **21.3265** |
| Memory as a Layer | 24.4822 |

Table 7: Comparison of VideoTitans performance with different memory integration strategies.

**Impact of Attention Mechanism** Table 6 compares the effect of different attention strategies—Sliding Window Attention, Global Attention, and no attention. Sliding Window Attention achieves the lowest MSE (21.3265), demonstrating its superior capability to effectively focus on local temporal dynamics compared to Global Attention (MSE: 24.4822) or removing attention entirely (MSE: 30.79).

**Memory Integration Strategies** Table 7 compares the performance of VideoTitans on MMNIST using three different memory integration strategies of Titans [2]: Memory as a Context (MAC), Memory as a Gate (MAG), and Memory as a Layer (MAL). MAG achieves significantly better performance (lowest MSE), clearly outperforming both MAC and MAL. This demonstrates that incorporating memory using a gating mechanism is particularly effective at capturing and selectively integrating critical historical information, leading to superior video prediction results.

# 6 Limitation and Future Work

In this work, we extensively evaluate VideoTitans on the video prediction task. Although our proposed model demonstrates strong generalization across multiple diverse datasets, further studies should investigate the effectiveness of our method in broader vision tasks such as action recognition, video segmentation, and anomaly detection.

While diffusion-based models have recently shown strong performance in video generation, our focus is on video prediction—forecasting future frames based on observed inputs, rather than generating from noise or prompts. Diffusion models involve iterative denoising processes with significant computational cost, prioritizing high-quality synthesis. In contrast, video prediction demands temporal consistency and real-time efficiency. Therefore, we compare against models specifically designed for future frame prediction. Still, leveraging the high fidelity of diffusion models alongside the real-time efficiency of predictive models offers a promising direction for long-range video forecasting.

Finally, we observe that despite the compelling strengths of the Titans concept, the choice of hyperparameters significantly impacts performance stability. Particularly, the neural memory module depth and gradient constraints are crucial; exceeding two layers frequently causes numerical instability during training. One important contribution of this work is the identification and documentation of these critical hyperparameters (Table 4), enabling stable and reproducible implementations of VideoTitans in future research.

# 7 Conclusion

In this work, we propose VideoTitans, a neural architecture designed for spatio-temporal video prediction, effectively capturing both local motion dynamics and long-term dependencies. By integrating three key components—Short-Term Memory with attention-based processing for recent frames, Long-Term Memory for selectively encoding and retrieving historical contexts, and Persistent Memory for task-specific knowledge—VideoTitans efficiently models complex video sequences while maintaining computational scalability. Extensive experimental evaluations demonstrate that VideoTitans consistently outperforms CNN-based, Transformer-based, and recurrent models, achieving superior predictive accuracy while significantly improving efficiency for long-term forecasting. These results underscore the effectiveness of VideoTitans as a robust and scalable solution for video-based predictive modeling, paving the way for advancements in real-world applications such as autonomous systems, surveillance, and robotics.

**Acknowledgement.** This work was partially supported by the National Research Foundation of Korea(NRF) grant funded by the Korea government(MSIT)(RS-2024-00338439) and Institute for Information & Communications Technology Planning & Evaluation (IITP) grant funded by the Korea government(MSIT) (No.RS-2025-25441838, Development of a human foundation model for human-centric universal artificial intelligence and training of personnel).

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

# A  Additional Dataset

## A.1  Caltech-Pedestrian Dataset

The Caltech-Pedestrian [14] dataset presents challenging real-world urban scenarios involving diverse pedestrian movements, occlusions, and complex dynamics. It is evaluated using metrics such as Mean Square Error (MSE), Structural Similarity Index Measure (SSIM), Peak Signal-to-Noise Ratio (PSNR), and Learned Perceptual Image Patch Similarity (LPIPS)[75], testing the robustness and accuracy of predictive models. As shown in Table8, VideoTitans demonstrates competitive performance across these metrics, effectively capturing intricate pedestrian trajectories and spatial relationships. This highlights its practical applicability in dynamic environments, combining predictive accuracy with computational efficiency.

| Method | MSE (↓) | SSIM (↑) | PSNR (↑) | LPIPS (↓) | FLOPs(G) (↓) |
|---|---|---|---|---|---|
| ConvLSTM | 139.6588 | 0.9345 | 27.4644 | 0.0857 | 595.0 |
| E3D-LSTM | 199.1374 | 0.9047 | 25.4612 | 0.1261 | 1004.0 |
| MIM | **123.9034** | 0.9410 | 28.1148 | 0.0642 | 1858.0 |
| PhyDNet | 310.6844 | 0.8615 | 23.2723 | 0.3218 | 40.4 |
| PredRNN | 129.3306 | 0.9375 | 27.8074 | 0.0745 | 1216.0 |
| PredRNNv2 | 143.4366 | 0.9334 | 27.1864 | 0.0895 | 1223.0 |
| MAU | 177.4630 | 0.9174 | 26.1504 | 0.0969 | 172.0 |
| TAU | 128.9193 | **0.9458** | 27.8465 | 0.0551 | 80.0 |
| SimVP-IncepU | 160.2191 | 0.9338 | 26.8093 | 0.0675 | 60.6 |
| SimVP-gSTA | 127.7992 | 0.9456 | 27.9191 | 0.0577 | 96.3 |
| SimVP-Swin | 155.2470 | 0.9300 | 27.2542 | 0.0811 | 95.2 |
| SimVP-Uniformer | 135.9496 | 0.9393 | 27.6607 | 0.0687 | 104.0 |
| SimVP-ViT | 146.3816 | 0.9380 | 27.4267 | 0.0666 | 155.0 |
| SimVP-Poolformer | 153.3675 | 0.9334 | 27.3807 | 0.0700 | 79.8 |
| VideoTitans | 130.4290 | 0.9448 | **28.8861** | **0.0512** | **9.9** |

Table 8: Performance comparison on Caltech Pedestrian dataset.

## A.2  KTH Dataset

The KTH [49] dataset is characterized by structured human actions and stable motion patterns which test a model's ability to capture temporal dynamics and spatial coherence in controlled settings. As shown in Table 9 VideoTitans achieves state-of-the-art performance across all evaluation metrics including MSE, Mean Absolute Error (MAE), PSNR, and SSIM. It combines high predictive accuracy with low computational complexity which confirms its practical effectiveness and shows its strength in modeling regular motion sequences with precision and efficiency.

| Method | MSE (↓) | MAE (↓) | PSNR (↑) | SSIM (↑) | FLOPs(G) (↓) |
|---|---|---|---|---|---|
| ConvLSTM | 47.65 | 445.5 | 26.99 | 0.8977 | 1368.0 |
| E3D-LSTM | 136.40 | 892.7 | 21.78 | 0.8153 | 217.0 |
| MIM | 40.73 | 380.8 | 27.78 | 0.9025 | 1099.0 |
| PhyDNet | 91.12 | 765.6 | 23.41 | 0.8322 | 93.6 |
| PredRNN | 41.07 | 380.6 | 27.95 | 0.9097 | 2800.0 |
| PredRNNv2 | 39.57 | 368.8 | 28.01 | 0.9099 | 2815.0 |
| MAU | 51.02 | 471.2 | 26.73 | 0.8945 | 399.0 |
| TAU | 45.32 | 421.7 | 27.10 | 0.9086 | 73.8 |
| SimVP-IncepU | 41.11 | 397.1 | 27.46 | 0.9065 | 62.8 |
| SimVP-gSTA | 45.02 | 417.8 | 27.04 | 0.9049 | 76.8 |
| SimVP-Swin | 45.72 | 405.7 | 27.01 | 0.9039 | 75.9 |
| SimVP-Uniformer | 44.71 | 404.6 | 27.16 | 0.9058 | 78.3 |
| SimVP-ViT | 56.57 | 459.3 | 26.19 | 0.8947 | 112.0 |
| SimVP-Poolformer | 45.55 | 400.9 | 27.22 | 0.9065 | 63.6 |
| VideoTitans | **34.27** | **320.8** | **29.31** | **0.9197** | **50.9** |

Table 9: Performance comparison on KTH dataset.

# B Qualitative Results

Figure 6 shows qualitative comparisons between VideoTitans, recurrent (PredRNNv2), and transformer-based (ViT) methods on the Moving MNIST dataset. Due to the dataset's relatively simple dynamics, all models perform similarly well, making it challenging to visually distinguish significant differences among predictions. Empirically, we observe that differences primarily lie in convergence speed rather than final performance, as extending training epochs tends to improve accuracy for all models. Nevertheless, VideoTitans consistently provides slightly more stable and accurate results. We also present qualitative results for t2m and uv10 variables from the WeatherBench dataset. Further qualitative results of VideoTitans are also available as GIF animations for better visualization.

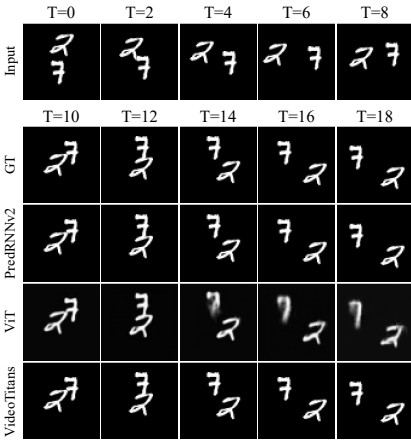

Figure 6: Qualitative comparison of predicted frames on the Moving MNIST dataset.

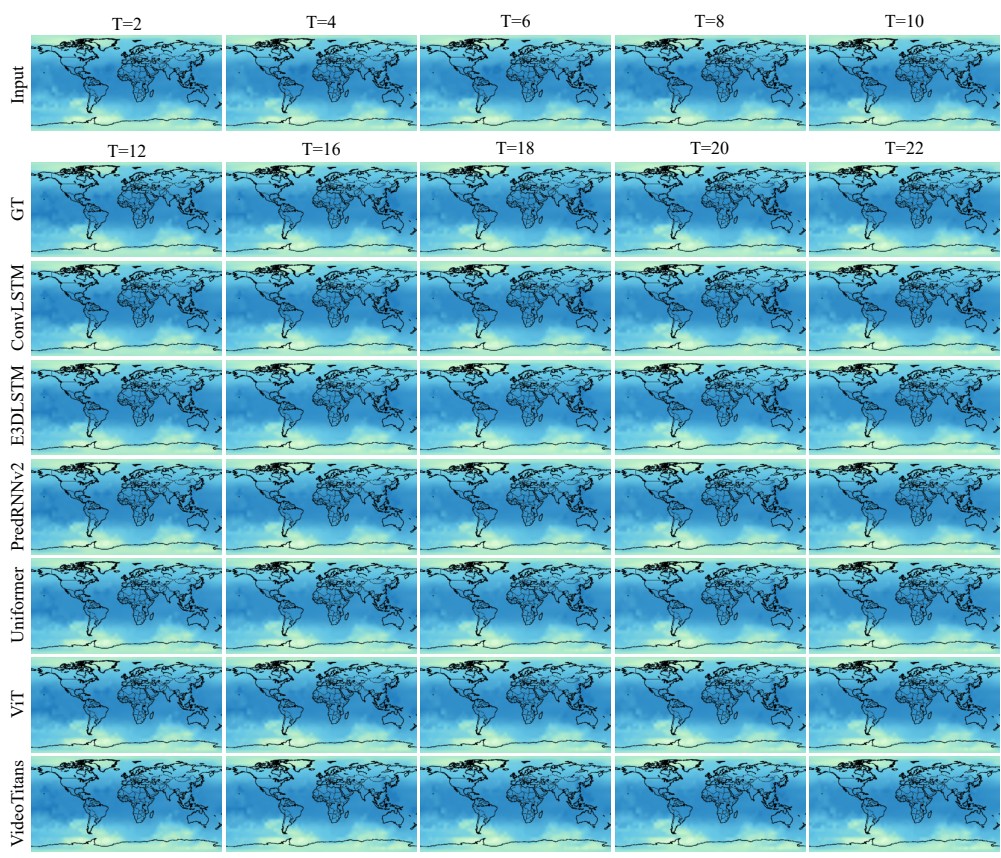

Figure 7: A qualitative comparison of predicted 2m temperature (`t2m`) frames on the WeatherBench dataset, comparing VideoTitans' predictions with those of other video prediction models.

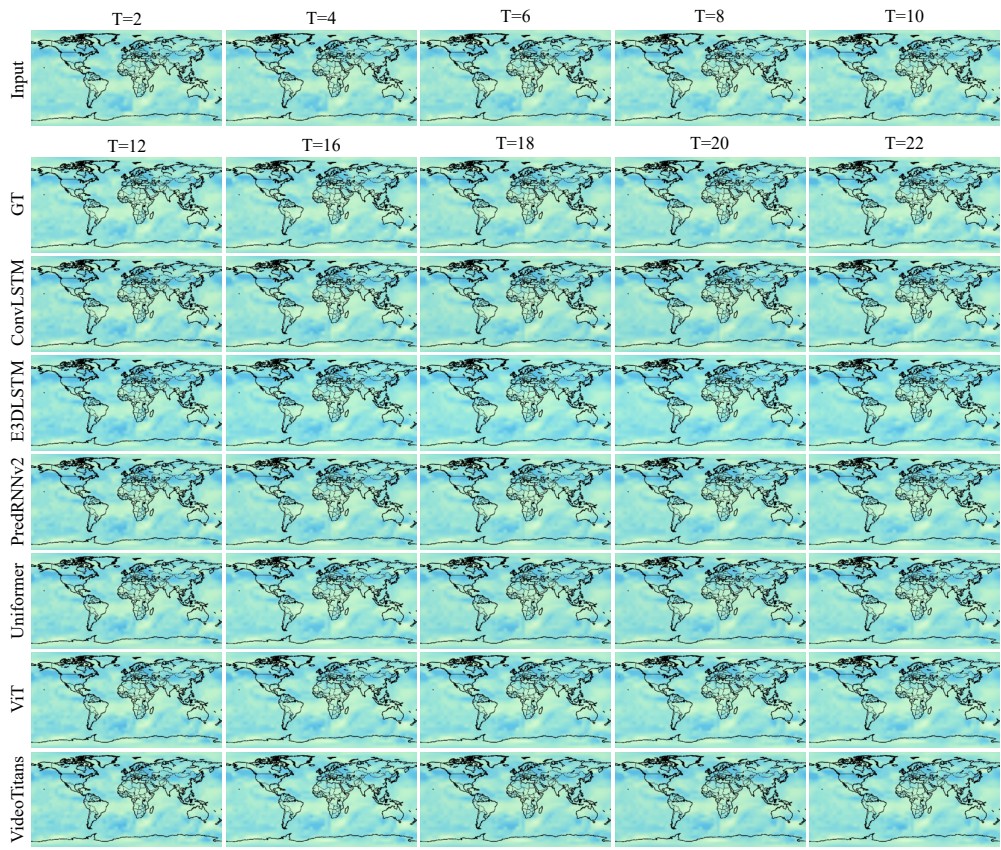

Figure 8: A qualitative comparison of predicted wind field (uv10) frames on the WeatherBench dataset, where VideoTitans' predictions are compared with those of other video prediction models.

## C   Implementation Details

### C.1   Model Architecture

The architecture of VideoTitans consists of three main components: an encoder for spatial feature extraction, a Titans-based temporal modeling module, and a decoder for frame reconstruction.

**Encoder.**   The encoder captures spatial and low-level visual features from input frames. Given an input tensor of shape $(B, T, C, H, W)$, each frame is independently processed by convolution-based patch embedding. Specifically, we employ a convolutional layer with kernel size $16 \times 16$ and stride 16, converting the input as:

$$(B \times T, C, H, W) \rightarrow (B \times T, \text{embed\_dim}, H/16, W/16).$$

Afterward, spatial positional encodings are added to preserve positional information. The tensor is then reshaped for temporal processing:

$$(B \times T, \text{embed\_dim}, H/16, W/16) \rightarrow (B, H/16 \times W/16, T \times \text{embed\_dim}).$$

**Titans-based Temporal Modeling.**   The temporal modeling is based on the Titans architecture, utilizing neural long-term memory that adaptively updates weights via a gradient-based surprise metric, efficiently capturing essential temporal patterns. Key hyperparameters, such as memory depth, memory dimension, persistent memory tokens, and maximum gradient norm, are critical for stable training. In particular, setting the maximum gradient norm to 1.0 prevents training instabilities such as gradient explosions.

The Titans module processes embeddings in segments, employing sliding window attention to model both local and global temporal dependencies. Persistent memory tokens encode context-independent knowledge to enhance generalization across datasets.

**Decoder.**   The decoder reconstructs predicted frames from the temporal features. Mirroring the encoder structure, it utilizes transpose convolutional layers (kernel size $16 \times 16$, stride 16) to restore spatial dimensions:

$$(B, H/16 \times W/16, T \times \text{embed\_dim}) \rightarrow (B \times T, C, H, W).$$

The decoded frames are reshaped to the original dimensions $(B, T, C, H, W)$ for comparison with ground-truth.

### C.2   Training Procedure

We implement VideoTitans in PyTorch, using the Adam optimizer and Mean Squared Error (MSE) loss function. Key training parameters are summarized below:

- **Optimizer:** Adam optimizer ($\beta_1 = 0.9$, $\beta_2 = 0.999$).
- **Learning Rate Scheduler:** ReduceLROnPlateau (patience=10 epochs), initial learning rate chosen from $\{10^{-2}, 5 \times 10^{-3}, 10^{-3}, 5 \times 10^{-4}, 10^{-4}\}$.
- **Batch Size:** 8 for all experiments.
- **Training Epochs:** MMNIST (200 epochs), Caltech Pedestrian (100 epochs), Human3.6, TrafficBJ, WeatherBench (50 epochs each).

Additionally, we apply the Exponential Moving Average (EMA) with a decay of 0.995 during training to enhance model stability and generalization.

### C.3   Hyperparameter Sensitivity

A key contribution of our study includes identifying sensitive hyperparameters essential for VideoTitans' stable training. Notably, removing gradient norm constraints (e.g., setting max gradient norm) caused training instabilities, and overly deep neural memory layers (depth > 2) frequently result in numerical instability. Careful hyperparameter tuning is thus essential for robust training and optimal performance.

## C.4 Memory Integration Strategies

There are three types of memory integration strategies in Titans: Memory as a Gate (MAG), Memory as a Context (MAC), and Memory as a Layer (MAL). MAG uses a gating mechanism to dynamically combine short-term attention and long-term memory, allowing the model to integrate previous knowledge adaptively. MAC retrieves past information from memory and appends it to the input sequence before processing it with attention, enabling selective use of historical data. MAL incorporates memory as an independent processing layer before the attention, similar to traditional hybrid recurrent models. Among these approaches, MAG achieves the best performance by effectively balancing short-term precision with long-term recall, leading to its selection as the baseline model for our work.

