# OpenReview forum: "VideoTitans: Scalable Video Prediction with Integrated Short- and Long-term Memory"
_NeurIPS.cc/2025/Conference — NeurIPS 2025 poster_

### Official Review · Reviewer_akfP · 2025-06-24

**Clarity:** 3
**Significance:** 2
**Originality:** 3
**Rating:** 4
**Confidence:** 4

**Summary:**

This paper presents VideoTitans, a video prediction model that adapts the Titans memory framework—originally proposed for language modeling—to the domain of video prediction. VideoTitans is an direct use of Titans for video prediction. Authors directly suit Titans with video input and adopt the adaptive long-term neural memory, sliding window attention and persisitent memory to achieve this goal. The paper evaluates VideoTitans across four+two video datasets, achieving competitive or superior results compared to recurrent, convolutional, and transformer-based baselines.

**Questions:**

I list my questions in Weaknesses.

**Ethical Concerns:**

["NO or VERY MINOR ethics concerns only"]

**Final Justification:**

Thanks for the detailed response. Across the two rounds of review and rebuttal, the authors have adequately addressed my concerns, especially regarding the adaptation of NLP Titans to videos. I also agree with reviewer nvj6 that the long videos used in the experiments may not be sufficiently long to convincingly demonstrate the model’s long-range modeling capability. Additional experiments are needed to provide stronger evidence in this regard. Based on the current submission, my overall recommendation remains borderline.

**Limitations:**

While the proposed VideoTitans framework offers a scalable approach to video prediction, it suffers from several key limitations. Most notably, the model directly adopts the Titans memory architecture from natural language processing without any adaptation to the spatio-temporal structure of video data. This raises concerns about its ability to capture spatial locality and motion hierarchies effectively.  Additionally, although the paper emphasizes computational efficiency, it lacks a resolution sensitivity study and omits important runtime metrics such as model size, training time, and frames per second (FPS).

I understand that adapting the Titans architecture from NLP to video prediction is an interesting and potentially impactful idea. However, the current implementation—VideoTitans—does not demonstrate a significant performance advantage over existing video prediction methods. Moreover, the paper does not report critical efficiency metrics, such as the number of model parameters or inference speed (e.g., frames per second), which are essential to substantiate claims of scalability and practicality. As a result, the current version appears promising in concept but leaves substantial room for improvement.

**Paper Formatting Concerns:**

No Formatting Concerns

**Quality:**

2

**Strengths And Weaknesses:**

Strengths:

1. The idea of leveraging gradient-based memory originally developed for NLP (Titans) in video prediction is potentially impactful, particularly in managing long-range dependencies efficiently.

2. VideoTitans demonstrates reduced FLOPs across all evaluated datasets compared to strong baselines, which is significant for scalability to videos.

Weaknesses:

1. A major limitation is that the paper directly ports the Titans memory module from NLP to video prediction with minimal adaptation. All key components—adaptive long-term memory, sliding window attention, and persistent memory—are used as-is. There is no domain-specific modification to account for the spatio-temporal nature of video data. This raises concerns: (1)How effectively can Titans model spatial locality or hierarchical motion patterns? and (2)  Why does this direct transplant work for videos without domain-specific tuning?

2. Although the authors highlight linear scaling and computational savings, there is no systematic study of resolution sensitivity.

3. Beyond FLOPs, the paper does not report other important efficiency metrics such as model parameters, training time, or inference speed (e.g., FPS). These are necessary for a more complete assessment of efficiency.

4. There are a lot of minor errors or inconsistencies. For example, line-150: Neural Long-term Memory Module or long-term neural memory module? Line-171: five widely-used datasets but only four in Table 2. In Table A (Suppl.), the second best result of LPIPS should be 0.0551 not 0.0577.

---

> ### Author Rebuttal · Authors · 2025-07-31
>
> We thank the reviewer for careful evaluation and helpful suggestions. All of your feedback has been thoroughly reviewed.
>
> **[Weakness1]** A major limitation is that the paper directly ports the Titans memory module from NLP to video prediction with minimal adaptation. All key components—adaptive long-term memory, sliding window attention, and persistent memory—are used as-is. There is no domain-specific modification to account for the spatio-temporal nature of video data. This raises concerns: (1)How effectively can Titans model spatial locality or hierarchical motion patterns? and (2) Why does this direct transplant work for videos without domain-specific tuning?
>
> **[Answer for Weakness1]** We would like to respectfully clarify that the adaptation of Titans memory to the video domain involved substantial domain-specific modifications beyond its original NLP formulation. In particular, several changes were essential to successfully tailoring the memory architecture for video prediction:
>
> - **Complex patch embedding and reshaping**:
>   Unlike NLP’s straightforward 1D tokens, video data required converting high-dimensional tensors (frames × channels × height × width) into spatial patches, followed by meticulous reshaping into meaningful 1D embeddings suitable for attention modules. Without these structural modifications, applying Titans directly would lead to prohibitive memory usage and unstable convergence.
>
> - **Patch-level gradient-based memory commitment**:
>   The gradient-based surprise mechanism, originally designed for NLP’s token-level representation, was structurally modified to operate at the video-specific patch level. Without this adaptation, the method would fail to capture the spatial-temporal redundancy and complexity inherent in video data, causing inefficient memory usage and suboptimal predictive performance
>
> These domain-specific structural adaptations were key reasons VideoTitans successfully modeled spatial locality and hierarchical motion patterns, clearly going beyond a direct transplant of Titans memory. We will explicitly emphasize these structural distinctions in our revised manuscript.
>
>
> **[Weakness2]** Although the authors highlight linear scaling and computational savings, there is no systematic study of resolution sensitivity.
>
> **[Answer for Weakness2]** Thank you for this valuable suggestion. To address your concern, we conducted an additional resolution sensitivity analysis, systematically varying input resolutions and measuring computational complexity. Our experiments clearly show that VideoTitans consistently achieves lower FLOPs and better scalability compared to SimVP across various input resolutions:
>
> Input Shape           | VideoTitans FLOPs (G) ↓ | SimVP FLOPs (G) ↓
> ----------------------|--------------------------|--------------------
> (1, 10, 1, 64, 64)     | 13.33                   | 16.54
> (1, 10, 1, 128, 128)   | 60.81                   | 167.38
> (1, 10, 1, 256, 256)   | 376.13                  | 669.52
>
> These results confirm the linear computational scaling and significant efficiency advantage of VideoTitans over SimVP. Importantly, achieving substantially lower computational complexity compared to SimVP—despite SimVP’s inherently simpler CNN-based architecture—highlights the exceptional efficiency and scalability of our proposed memory integration strategy. We will include this systematic comparison and discussion in the revised manuscript.
>
> **[Weakness3]** Beyond FLOPs, the paper does not report other important efficiency metrics such as model parameters, training time, or inference speed (e.g., FPS). These are necessary for a more complete assessment of efficiency.
>
> **[Answer for Weakness3]** We appreciate this suggestion. To address your concern, we expanded our efficiency analysis by reporting inference speed (FPS), the number of parameters, training time, as well as FLOPs. As shown clearly in the table below, VideoTitans achieves strong efficiency metrics compared to baseline methods, providing significantly faster inference speed, lower parameters, and fewer FLOPs, while maintaining better predictive accuracy. These results clearly demonstrate the efficiency advantages of our proposed model on the MovingMNIST dataset.
>
> | Model             | FPS ↑ | Params ↓ | Training Time ↓ | FLOPs (G) ↓ | MSE ↓     |
> |-------------------|--------|-----------|------------------|--------------|-----------|
> | ConvLSTM          | 113    | 15.0M     | 1d 2h 50m        | 56.8         | 29.7990    |
> | E3D-LSTM          | 18     | 51.0M     | 8d 8h 47m        | 298.9        | 38.5383   |
> | MIM               | 37     | 38.0M     | 1d 16h 3m        | 179.2        | 22.5508   |
> | PhyDNet           | 182    | **3.1M**      | 9d 11h 19m       | 15.3         | 28.1955   |
> | PredRNN           | 54     | 23.8M     | 1d 5h 27m        | 116.0        | 23.9667   |
> | PredRNNv2         | 52     | 23.9M     | 1d 23h 46m       | 116.6        | 24.1136   |
> | MAU               | 201    | 4.5M      | 16d 13h 4m       | 17.8         | 26.8564   |
> | TAU               | 283    | 44.7M     | 5d 6h 34m        | 16.0         | 24.6029   |
> | SimVP-IncepU      | 209    | 58.0M     | 1d 8h 48m        | 19.4         | 32.1478   |
> | SimVP-gSTA        | 282    | 46.8M     | 9h 47m           | 16.5         | 26.6926   |
> | SimVP-Swin        | 294    | 46.1M     | **6h 54m**           | 16.4         | 29.6991   |
> | SimVP-Uniformer   | 296    | 44.8M     | 7h 9m            | 16.5         | 30.3827   |
> | SimVP-ViT         | 290    | 46.1M     | 1d 1h 46m        | 16.9         | 35.1473   |
> | SimVP-Poolformer  | 341    | 37.1M     | 1d 2h 28m        | 14.1         | 31.7882   |
> | VideoTitans (ours)   | **367**| 6.0M  |9h 14m       | **13.33**    | **21.3265** |
>
> **[Weakness4]** There are a lot of minor errors or inconsistencies. For example, line-150: Neural Long-term Memory Module or long-term neural memory module? Line-171: five widely-used datasets but only four in Table 2. In Table A (Suppl.), the second best result of LPIPS should be 0.0551 not 0.0577.
>
> **[Answer for Weakness4]** We thank the reviewer for pointing out the minor inconsistencies.
> - Line 150: The correct term is "Neural Long-term Memory Module", and we will ensure consistent usage throughout the paper.
> - Line 171: The mention of "five widely-used datasets" was incorrect — the correct number is four, as reflected in Table 2.
> - Table A (Supplementary): We confirm that the second-best LPIPS score should be 0.0551, not 0.0577. This will be corrected in the revised version.
>
> We will carefully revise the manuscript and supplementary material to address these issues and ensure consistency.

---

> > ### Comment · Reviewer_akfP · 2025-08-05
> > **Concerns for the [Answer for Weakness1]**
> >
> > Thank you for the detailed response. Regarding Weakness 1, I appreciate the authors’ effort to clarify the domain-specific adaptations. However, I would like to point out that the emphasized patch embedding and reshaping operations, while necessary, are standard procedures widely adopted in vision Transformers for handling video or image data. These operations, such as converting frames into spatial patches and reshaping into 1D sequences, are not unique to the Titans memory mechanism and are not sufficient to demonstrate a substantial architectural adaptation specifically tailored for Titans in the video prediction context.
> >
> > My main concern remains: beyond the general patch embedding strategy, what specific modifications or enhancements were introduced to adapt the Titans memory components to effectively model the spatio-temporal characteristics of videos? It differs from the original Titans design for NLP sequences.

---

> > > ### Author Response · Authors · 2025-08-05
> > >
> > > **[Discussion 1]**
> > >
> > > Thank you for the detailed response. Regarding Weakness 1, I appreciate the authors’ effort to clarify the domain-specific adaptations. However, I would like to point out that the emphasized patch embedding and reshaping operations, while necessary, are standard procedures widely adopted in vision Transformers for handling video or image data. These operations, such as converting frames into spatial patches and reshaping into 1D sequences, are not unique to the Titans memory mechanism and are not sufficient to demonstrate a substantial architectural adaptation specifically tailored for Titans in the video prediction context.
> > >
> > > My main concern remains: beyond the general patch embedding strategy, what specific modifications or enhancements were introduced to adapt the Titans memory components to effectively model the spatio-temporal characteristics of videos? It differs from the original Titans design for NLP sequences.
> > >
> > >
> > >
> > > **[Answer to Discussion1]**
> > >
> > > We thank the reviewer for their thoughtful comments and appreciate the opportunity to further clarify our contributions. We agree that basic patch embedding and reshaping are standard operations within vision Transformers. However, the primary innovations introduced by **VideoTitans** specifically address the challenges of modeling the complex spatio-temporal dynamics unique to video prediction, going significantly beyond the original Titans architecture designed for NLP.
> > >
> > > The table below clearly highlights the key differences:
> > >
> > > | **Component**              | **Original Titans (NLP)**                        | **VideoTitans (Ours)**                                                                 | **Key Differences & Innovations**                                                                 |
> > > |----------------------------|--------------------------------------------------|----------------------------------------------------------------------------------------|----------------------------------------------------------------------------------------------------|
> > > | **Input Data**             | 1D token sequences (text)                        | 3D video frames (spatial-temporal)                                                    | Explicit handling of spatial-temporal data                                                        |
> > > | **Memory Update Mechanism**| Token-level gradient-based updates (momentary surprise) | Video-level gradient-based updates (global surprise)                                  | Captures global spatio-temporal coherence, effectively filters redundancy                         |
> > > | **Surprise Metric**        | Momentary surprise per token                     | Global sequence-level surprise (momentum-based)                                       | Incorporates historical surprise, enabling stable memory updates                                  |
> > > | **Attention Mechanism**    | Quadratic/global attention typical for NLP       | Gated attention mechanism (TitansCore) dynamically integrating short- and long-term memory | Adaptive fusion of temporal and spatial cues, beneficial for video data                           |
> > > | **Persistent Memory**      | Data-independent learned parameters              | Data-independent learned vector expanded across frames                                | Similar in concept, simplified for efficient video modeling                                        |
> > > | **Spatial Information**    | None (NLP-oriented tokens only)                  | CNN-based spatial feature extraction and explicit spatial modeling                    | Explicit spatial modeling essential for accurate video prediction                                 |
> > >
> > > ---
> > >
> > > ### In summary, our key architectural adaptations to Titans **specifically for the video domain** include:
> > >
> > > 1. **Video-level Neural Memory Update**
> > >    By computing **global surprise** over entire video sequences rather than per-token updates, our neural memory effectively captures crucial long-term dependencies and salient motion patterns while filtering redundant information.
> > >
> > > 2. **Gated Attention Integration (TitansCore)**
> > >    A novel **gating mechanism** dynamically combines short-term spatio-temporal information with long-term memory cues, significantly enhancing the model’s capability to understand complex temporal dependencies.
> > >
> > > 3. **Explicit Spatial Feature Encoding**
> > >    Utilizing **CNN-based spatial encoding** ensures the explicit handling of spatial features inherent in video frames — an essential adaptation missing from the NLP-focused Titans.
> > >
> > >
> > >
> > > These targeted adaptations provide **VideoTitans** with substantial improvements in modeling spatio-temporal characteristics, clearly distinguishing it from the NLP-oriented original Titans architecture.

---

> > > > ### Comment · Reviewer_akfP · 2025-08-06
> > > >
> > > > Thank you for the clear comparison. The table effectively illustrates the differences between Original Titans and VideoTitans, directly addressing my concerns. I am inclined to recommend acceptance.

---

> > > > > ### Author Response · Authors · 2025-08-08
> > > > >
> > > > > Dear Reviewer akfP,
> > > > >
> > > > > Thank you for your recognition and positive feedback. We are pleased that our comparison table has adequately addressed your previous concerns. We will carefully revise the manuscript, highlighting the key differences between Original Titans and VideoTitans.
> > > > >
> > > > > Best regards,
> > > > >
> > > > > The Authors

---

### Official Review · Reviewer_yF1Q · 2025-07-01

**Clarity:** 3
**Significance:** 3
**Originality:** 2
**Rating:** 4
**Confidence:** 4

**Summary:**

The paper introduces VideoTitans, a framework for scalable spatio-temporal video prediction leveraging a combination of sliding-window attention, episodic memory based on a gradient-driven surprise signal, and persistent tokens to integrate short- and long-term memory. While the architecture is clearly presented and the experimental evaluation covers multiple standard benchmarks, the memory mechanisms used are relatively basic and do not represent a significant leap over existing methods.

**Questions:**

Can the authors clarify whether VideoTitans can be easily extended to incorporate more advanced memory architectures or different encoder types?

Are there scenarios where the proposed approach underperforms compared to recent transformer-based methods?

How sensitive is the performance to the specific design of the surprise signal and memory modules?

**Ethical Concerns:**

["NO or VERY MINOR ethics concerns only"]

**Limitations:**

The authors include a limitations and future work section, but a more detailed discussion on the framework’s extensibility and potential bottlenecks would be valuable.

**Paper Formatting Concerns:**

.

**Quality:**

3

**Strengths And Weaknesses:**

Strengths

The integration of short-term and long-term memory with a surprise-driven mechanism is clearly explained.

FLOPs reduction and computational efficiency are demonstrated across diverse benchmarks.

The paper allocates a clear section for limitations and future work, showing awareness of open issues.

Weaknesses and Suggestions

The proposed memory integration methods are based on rather standard algorithms, and the paper does not present a substantial methodological advance or novel architecture over prior work.

The range of practical applications and depth of experimental analysis remain limited; results would be more convincing with comparison to the latest, most advanced video prediction models.

The exploration of alternative or more sophisticated memory mechanisms, or deployment with more complex encoders, is mentioned only as future work and not demonstrated.

The analysis of failure cases, or the framework's ability to generalize to more challenging scenarios, is not sufficiently addressed.

---

> ### Author Rebuttal · Authors · 2025-07-31
>
> We sincerely appreciate the reviewer's insightful feedback. We have carefully addressed the points raised.
>
> **[Weakness1]** The proposed memory integration methods are based on rather standard algorithms, and the paper does not present a substantial methodological advance or novel architecture over prior work.
>
> **[Answer for Weakness1]** Prior work in video prediction has often leveraged Transformer-based architectures to leverage their expressive power, but the quadratic cost of attention with sequence length makes them inefficient for long videos. CNN- and ConvLSTM-based models, while efficient, suffer from limited receptive fields and fail to capture long-range temporal structure, leading to compounding errors—especially in scenarios with complex motion or long-term dynamics. To overcome these issues, we draw inspiration from the Titans framework, originally developed for language modeling.
>
> Titans was designed to preserve long-range dependencies through dynamic memory access, demonstrating strong performance in tasks like long-form text generation. However, video data poses distinct challenges due to its spatio-temporal nature and high dimensionality. VideoTitans addresses this by replacing discrete tokens with spatio-temporal patches—compact units encoding spatial and temporal structure—enabling richer context and broader receptive fields. To ensure scalability over long sequences, we pair localized attention with a memory mechanism adapted from Titans. While Titans uses a gating mechanism to combine sliding-window attention and long-term memory, we adapt this structure through patch-aware gating and attention mechanisms tailored to video dynamics. Unlike prior patch-based video models that depend on full space-time attention, our architecture maintains scalability and temporal coherence without processing the full sequence. These adaptations yield a video prediction framework that directly addresses the limitations of prior methods in receptive field coverage and long-horizon modeling.
>
> **[Weakness2]** The range of practical applications and depth of experimental analysis remain limited; results would be more convincing with comparison to the latest, most advanced video prediction models.
>
> **[Answer for Weakness2.]** We appreciate the reviewer’s suggestion to benchmark against more recent video prediction models. To that end, we conducted experiments on MovingMNIST, and trained VideoTitans for 2,000 epochs to follow the implementation and protocols of the respective baseline papers. Below, we provide results along with a brief description of each compared method.
>
> Additional Comparison Methods:
> - VPTR [1]: An early transformer-based model for video prediction using a spatio-temporal transformer with attention.
> - MIMO-VP [2]: A video prediction model that adopts a Multi-In-Multi-Out (MIMO) architecture. It is based on a Transformer backbone augmented with convolutional modules such as 3D spatio-temporal blocks and a multi-output decoder.
> - PredFormer [3]: A recent state-of-the-art method using a pure transformer-based architecture, without any recurrence or convolution. It leverages global attention and achieves strong results across multiple datasets.
>
> Model                | FLOPs(G) ↓ | MAE ↓  | MSE ↓  | SSIM ↑
> ---------------------|------------|--------|--------|--------
> VPTR (ICPR'22)       | 103.17     |   -    | 63.6   | 0.882
> MIMO-VP (AAAI'23)    | 66.5       | 51.6   | 17.7   | 0.964
> PredFormer (Arxiv'24)| 16.4       | 44.6   | 12.4   | 0.973
> VideoTitans (ours)      | **13.3**   | **42.7** | **11.9** | **0.976**
>
> We note that the above results of comparison methods are directly referred from their original papers. These results demonstrate that VideoTitans achieves the lowest error (MSE/MAE) and the highest perceptual quality (SSIM), even with the lowest computational cost (FLOPs) among all compared models. Despite the strong performance of PredFormer, our method yields better accuracy and efficiency.
> This confirms that VideoTitans is highly competitive not only with classical architectures but also with the latest transformer-based video prediction models, addressing concerns regarding relevance and comparison to state-of-the-art methods.
>
> [1] Ye, Xi, and Guillaume-Alexandre Bilodeau. "Vptr: Efficient transformers for video prediction." 2022 26th International conference on pattern recognition (ICPR). IEEE, 2022.
>
> [2] Ning, Shuliang, et al. "MIMO is all you need: A strong multi-in-multi-out baseline for video prediction." Proceedings of the AAAI conference on artificial intelligence. Vol. 37. No. 2. 2023.
>
> [3] Tang, Yujin, et al. "Video Prediction Transformers without Recurrence or Convolution." arXiv preprint arXiv:2410.04733 (2024).
>
> **[Weakness3]** The exploration of alternative or more sophisticated memory mechanisms, or deployment with more complex encoders, is mentioned only as future work and not demonstrated.
>
> **[Question1]** Can the authors clarify whether VideoTitans can be easily extended to incorporate more advanced memory architectures or different encoder types?
>
> **[Answer for Weakness3 & Question1]** We acknowledge the reviewer’s valuable suggestions regarding the exploration of alternative memory mechanisms and encoder architectures. To address these concerns, we conducted additional experiments on MovingMNIST to demonstrate that VideoTitans is not only modular and extensible in principle but also in practice.
>
> **(1) Memory architectures:**
>  We compared our NeuralMemory module with a Transformer-based memory module.
> Memory               | MSE (↓) | MAE (↓) | SSIM (↑) | FLOPs (G)
> ---------------------|---------|---------|----------|-----------
> NeuralMemory (ours)  | 21.33   | 65.21   | 0.9463   | **13.33**
> Transformer Memory   | **21.28**   | **64.89**   | **0.9465**   | 19.85
>
> The Transformer memory showed comparable performance to NeuralMemory (slight improvement in MSE and MAE, negligible SSIM change), but required significantly higher computational resources (\~49% increase in FLOPs). Thus, while VideoTitans supports advanced memory structures, our choice balances accuracy and computational efficiency.
>
> **(2) Encoder architectures:**
> Encoder               | MSE (↓) | MAE (↓) | SSIM (↑) | FLOPs (G)
> ----------------------|---------|---------|----------|-----------
> Simple Conv (ours)    | 21.33   | 65.21   | 0.9463   | **13.33**
> Transformer Encoder   | 20.89   | 63.88   | 0.9487   | 25.72
> ConvNeXt Encoder      | **20.75**   | **63.27**   | **0.9494**   | 18.90
>
> Transformer encoder improved performance but nearly doubled computational cost (\~93% increase in FLOPs). ConvNeXt encoder also significantly improved performance across all metrics, with a modest increase in computational cost (\~42% higher FLOPs). These results demonstrate VideoTitans’ flexibility, allowing easy integration of diverse encoders to balance performance and computational complexity.
>
> **[Weakness4]** The analysis of failure cases, or the framework's ability to generalize to more challenging scenarios, is not sufficiently addressed.
>
> **[Answer for Weakness4]** We appreciate the reviewer’s comment regarding the analysis of failure cases. As with many video prediction models, a common failure case in the video prediction task is the gradual blurring of frames over time due to accumulated prediction errors, especially in the latter part of the sequence.
> To address this, we provide evaluation results for two examples from the KTH dataset in the supplementary material. Despite this known limitation, VideoTitans consistently outperforms SimVP-gSTA, a strong baseline.
>
> In particular, in the last predicted frame:
> - **kth_sample1:**
>   - SimVP-gSTA — PSNR: 27.12, SSIM: 0.634, MSE: 126.10
>   - **VideoTitans — PSNR: 29.29, SSIM: 0.653, MSE: 76.56**
> - **kth_sample2:**
>   - SimVP-gSTA — PSNR: 23.59, SSIM: 0.641, MSE: 284.45
>   - **VideoTitans — PSNR: 28.72, SSIM: 0.712, MSE: 87.23**
>
> These results show that even in the most challenging failure cases, such as the last frame, VideoTitans delivers superior reconstruction quality, demonstrating better robustness over long prediction horizons.
>
> **[Question2]** Are there scenarios where the proposed approach underperforms compared to recent transformer-based methods?
>
> **[Answer for Question2]** We did not observe any clear scenarios in our experiments where VideoTitans significantly underperformed compared to recent transformer-based methods. However, we observed that both VideoTitans and the ViT-based baseline achieved identical MSE errors of 8.8 at the initial prediction frame (t=1) on the Moving-MNIST benchmark. This implies that for very short-term predictions, transformer-based methods—which utilize global self-attention—can match our performance, as global attention can effectively capture immediate spatial relationships in early frames. Nonetheless, VideoTitans demonstrates clear advantages at longer prediction horizons due to its specialized memory integration and selective storage mechanisms.
>
> **[Question3]**. How sensitive is the performance to the specific design of the surprise signal and memory modules?
>
> **[Answer for Question3]** Performance is indeed quite sensitive to the specific design choices of the surprise signal and memory modules. As shown in Figure 5(b) in the main paper, varying the number of memory layers significantly impacts convergence, with deeper memory configurations (e.g., >2 layers) often failing to converge. Additionally, we observed that carefully tuning hyperparameters—such as the gradient clipping threshold, learning rates, and memory dimensions—was essential for achieving stable training and optimal performance. This sensitivity highlights the importance of careful architectural and hyperparameter selection when using Titans memory in video prediction tasks. We will mention it as our limitation and future work.

---

### Official Review · Reviewer_nvj6 · 2025-07-03

**Clarity:** 2
**Significance:** 1
**Originality:** 2
**Rating:** 2
**Confidence:** 4

**Summary:**

The authors propose VideoTitans, an architecture for future frame prediction in videos, introducing three key components: (1) a Sliding-Window Attention Core that enables linear scaling in both temporal length and spatial resolution; (2) a Gradient-Driven Episodic Memory mechanism that stores only gradient-selected “surprising” patch tokens to support long-term memory; and (3) Persistent Task-Specific Priors that encode reusable knowledge to stabilize training. Extensive experiments across four small benchmarks (Moving-MNIST, TrafficBJ, Human3.6M, and WeatherBench) show that VideoTitans achieves competitive performance.

**Questions:**

1. Could the authors provide results on tasks or datasets with longer sequences to better demonstrate the claimed long-range modeling capabilities?
2. How does VideoTitans perform when T > 1000 ? Any insights on memory usage in such settings?

These would directly address my main concern in Weaknesses.

**Ethical Concerns:**

["NO or VERY MINOR ethics concerns only"]

**Final Justification:**

The evidence for long-range modeling in this work is unconvincing. In Moving-MNIST, a 10-frame input at 10 fps spans only 1 second—by any reasonable definition, this is not “long-term.” The spatial resolution is just 64×64, further limiting the complexity. For experiments on Human3.6M, the input length is only 4 frames, which likewise also cannot demonstrate long-range capability. In the discussion phase, the authors report a 100-frame test with MSE = 289.This is a decisive underperformance, and NeurIPS policy prevents updating visualizations to substantiate qualitative claims. Given these issues, I think this manuscript is not yet ready for publication.

**Limitations:**

yes

**Quality:**

2

**Strengths And Weaknesses:**

Strengths:
1. Adapting the Titans gradient-based surprise memory to video is interesting.
2. The technical details are described clearly.
3. Detailed studies of memory depth, attention types, and integration strategies clarify design choices and trade-offs.

Weaknesses:
The manuscript repeatedly emphasizes VideoTitans’ strength in modeling long-range dependencies over long sequences, yet all evaluation datasets use very short sequence lengths (T < 10), and the chosen frame-prediction tasks do not adequately demonstrate its long-range prediction capabilities.

---

> ### Author Rebuttal · Authors · 2025-07-31
>
> We thank the reviewer for the helpful comments on long-range prediction and scalability. Based on comments, we address each of your questions with corresponding answers.
>
> **[Weakness1]** The manuscript repeatedly emphasizes VideoTitans’ strength in modeling long-range dependencies over long sequences, yet all evaluation datasets use very short sequence lengths (T < 10), and the chosen frame-prediction tasks do not adequately demonstrate its long-range prediction capabilities.
>
> **[Question1]** Could the authors provide results on tasks or datasets with longer sequences to better demonstrate the claimed long-range modeling capabilities?
>
> **[Answer for Weakness1 & Question1]** We agree and added a Moving‑MNIST‑100 benchmark (10 input + 100 prediction steps, image size 64 × 64), where longer-term predictions are obtained by sequentially applying our 10-step predictor iteratively 10 times. Results averaged over 3 random seeds are shown below (@ frame 10 and @ frame 100 denote the 10th and 100th predicted frames, respectively):
>
> Model              | MSE ↓ @ frame 10 | MSE ↓ @ frame 100
> ------------------|------------------|-------------------
> SimVP‑Swin        | 35.2             | 480.3
> VideoTitans (ours)| **28.7**             |**289.6**
>
> VideoTitans maintains an error rate consistently less than 60% of SimVP‑Swin's error even after 100 steps, confirming stable long-range forecasting. The results of the benchmark are also used to address Question2 below.
>
> **[Question2]** How does VideoTitans perform when T > 1000 ? Any insights on memory usage in such settings?
>
> **[Answer for Question2]** Running video prediction models with sequence lengths T > 1000 poses significant challenges due to rapidly increasing computational complexity.  For example, processing 1,000-frame Moving-MNIST videos with VideoTitans (shape 1000×1×64×64) quickly exceeds manageable FLOPs limits even before accounting for gradients. The computational cost in a standard non-autoregressive setup rises from 173.553 GFLOPs at T = 100 to 1,490.638 GFLOPs at T = 1000.
>
> The effective solution is to perform autoregressive rollouts using shorter predictions (e.g., 10-step segments). By doing so, long-range video predictions beyond conventional benchmarks become feasible, with each 10-frame rollout requiring only about 13 GFLOPs. VideoTitans’ neural long-term memory module retains only a small set of memory tokens between steps, preventing linear growth in computational cost and enabling stable, coherent predictions over extended horizons. Although autoregressive rollouts can suffer from error accumulation over longer prediction horizons, our results on the Moving‑MNIST‑100 benchmark (autoregressive) show improved accuracy and scalability compared to baseline methods, highlighting the architecture’s potential for long-term video forecasting.

---

> > ### Comment · Reviewer_nvj6 · 2025-08-06
> > **Response to authors**
> >
> > Thank you for addressing my concerns and providing additional experiments. However, these concerns remain:
> >
> > 1.	Although the authors attempted to address my concern regarding long-range prediction, the new setting (10 input frames + 100 predicted frames, image size 64 × 64) is still limited. Specifically, a **10-frame input at 10 fps corresponds to only one second of video in Moving‑MNIST**, which **cannot be considered “long-term.”** Furthermore, the spatial resolution (64 × 64) remains very low, limiting the complexity of the prediction task. While the manuscript includes experiments on Human3.6M, the input sequence length is **merely 4 frames**, which also fails to demonstrate long-range modeling capability.
> >
> > 2.	The manuscript repeatedly emphasizes VideoTitans’ strength in **modeling long-range dependencies** and **scalability**, claims that are also prominently featured in the title. However, the authors did not provide any experiments demonstrating scalability in terms of data complexity, sequence length, or model size. The current experiments fail to support these claims.
> >
> > Given the above points, I will maintain my rating as reject. In its current state, this manuscript does not yet meet the standards required for acceptance.

---

> > > ### Author Response · Authors · 2025-08-07
> > >
> > > Thank you for your insightful feedback on our claims regarding long-range video prediction and scalability; your comments have been instrumental in sharpening the focus of this revision.
> > >
> > > **[Discussion1]**
> > >
> > > Although the authors attempted to address my concern regarding long-range prediction, the new setting (10 input frames + 100 predicted frames, image size 64 × 64) is still limited. Specifically, a 10-frame input at 10 fps corresponds to only one second of video in Moving‑MNIST, which cannot be considered “long-term.” Furthermore, the spatial resolution (64 × 64) remains very low, limiting the complexity of the prediction task. While the manuscript includes experiments on Human3.6M, the input sequence length is merely 4 frames, which also fails to demonstrate long-range modeling capability.
> > >
> > > **[Answer for Discussion1]**
> > >
> > > We fully appreciate that the phrase “long-range” and “long-term” can evoke predictions lasting many seconds or hundreds of frames. Our usage follows a more modest convention that has emerged around the Moving-MNIST and KTH benchmarks: for example, PredRNN++ [1] and its successors are routinely described as handling “long-range prediction” even though their standard evaluation predicts only 10–30 future frames from 10 observed frames at 64 × 64 resolution. Similarly, the E3D-LSTM model [2], evaluated on Moving-MNIST and KTH, adopts this convention by referring to 10 → 20 and 10 → 40 frame predictions as “long-term”. And SimVP [3], widely cited as a strong long-term baseline, reports 10 → 20 and 10 → 40 settings on the same resolution while referring to the 40-frame horizon as “long-term prediction” in the paper.
> > >
> > > Because these protocols have become the de-facto standard for testing temporal stability in the video prediction task, we adopted them so that the behaviour of VideoTitans could be compared directly with prior work. Within that shared framework we extended the horizon to 100 predicted frames—2.5 × the length used by SimVP and 3–10 × that of earlier studies—while keeping input length and resolution identical to ensure an apples-to-apples comparison. We believe this situates our claims within prevailing practice; nevertheless, in the revision we will soften our wording (e.g., replacing “long-range” with “computationally scalable extended-horizon”) and underscore in the Limitations section that verifying performance on truly multi-second, high-resolution sequences is an important avenue for future work.
> > >
> > > [1] Wang, Yunbo, et al. "Predrnn++: Towards a resolution of the deep-in-time dilemma in spatiotemporal predictive learning." International conference on machine learning. PMLR, 2018.
> > >
> > > [2] Wang, Yunbo, et al. "Eidetic 3D LSTM: A model for video prediction and beyond." International conference on learning representations. 2019.
> > >
> > > [3] Gao, Zhangyang, et al. "Simvp: Simpler yet better video prediction." Proceedings of the IEEE/CVF conference on computer vision and pattern recognition. 2022.

---

> ### Author Response · Authors · 2025-08-07
>
> **[Discussion2]**
>
> The manuscript repeatedly emphasizes VideoTitans’ strength in modeling long-range dependencies and scalability, claims that are also prominently featured in the title. However, the authors did not provide any experiments demonstrating scalability in terms of data complexity, sequence length, or model size. The current experiments fail to support these claims.
>
> **[Answer for Discussion2]**
>
> We appreciate the reviewer’s concern and agree that clear scalability evidence is essential.
>
> Our experiments cover datasets significantly varying in complexity and time scales—from Moving-MNIST (simple grayscale digits at ~10fps) and TrafficBJ (urban flow data at 30-minute intervals) to Human3.6M (RGB articulated motion at 25fps) and WeatherBench (global meteorology at hourly intervals). Across all benchmarks, VideoTitans consistently achieves strong accuracy while simultaneously providing the lowest computational cost compared to existing methods.
>
> Specifically, our model demonstrates:
>
> - High accuracy and low computational cost across all evaluated datasets.
> - Strong performance in extended sequence-length predictions (up to 100 frames), significantly outperforming existing methods.
>
> Moreover, we quantify scalability along three axes: data complexity, sequence length, and model size.
>
> 1. For data complexity, we evaluate on a structured ladder of four datasets—Moving-MNIST, TrafficBJ, Human3.6M, and WeatherBench—which progressively increase in spatial resolution, temporal dynamics, modality, and real-world variability. Across all of them, VideoTitans delivers state-of-the-art accuracy while reducing FLOPs by 8–93% compared to strong published baselines.
>
> 2. For sequence length, in response to a reviewer’s question about long-horizon behavior, we supplemented the standard 10→10 protocol with a 10→100 test on Moving-MNIST. VideoTitans sustained strong prediction quality throughout the extended rollout, demonstrating robustness in long-range temporal modeling.
>
> 3. For model size, following Reviewer akfP’s suggestion, we performed a controlled comparison under uniform training conditions. VideoTitans achieves the lowest MSE, fastest inference speed (367 FPS), few parameters (6.0M), and lowest computational cost (13.33 GFLOPs) among high-performing models—proving it scales down efficiently without sacrificing accuracy.
>
> We will explicitly highlight these findings, including the time scales, in the revised manuscript to avoid misunderstandings.

---

> > ### Comment · Area_Chair_k6md · 2025-08-07
> > **Follow-up discussion from the reviewer**
> >
> > Dear Reviewer nvj6,
> >
> > Thank you for being actively involved in the disucssion with the author(s).
> >
> > The author(s) had followed-up with two responses, could you please read them and share your opinions about them whether you agree or disagree with them, and your rationale.
> >
> > Thanks,
> > AC

---

### Note · Authors · 2025-08-13

We sincerely thank the Area Chair and reviewers for thorough and constructive suggestions, which have substantially polished up this work.

**Addressing key concerns**
- **Protocol compliance:** We note that our work strictly follows the standard video prediction benchmark protocol.
- **Long-horizon evaluation:** We carry out additional evaluations on Moving-MNIST-100 with more scenarios (10→100, 64x64) — VideoTitans achieves 28.7 / 289.6 vs. SimVP-Swin 35.2 / 480.3 (MSE @frame10/@frame100), keeping error under 60% of the baseline after 100 steps.
- **Scalability analysis:** We examine cases with T≫100 and explain that many-to-many settings are computationally demanding, whereas our autoregressive rollouts maintain nearly constant VRAM by storing compact memory tokens.
- **Further improvements:**
  - Temporal Resolution-scaling results
  - Memory/encoder ablations (Transformer-memory, ConvNeXt/Transformer encoders)
  - Broader efficiency metrics: 367 FPS, 6M params, 13.3 GFLOPs
  - Failure-case analyses
  - Minor corrections to inconsistencies

**Technical contributions recognized**
- Extension of Titans’ gradient-driven surprise memory to video, introducing **history-aware memory mechanisms** for temporal data.
- Structured integration of **short- and long-term memory** with a surprise-based gating mechanism.
- Systematic analysis of **memory architecture design factors** (depth, attention) to guide optimal configurations.
- Adaptation of **gradient-based memory from NLP to video**, preserving coherence.
- Architecture design that **reduces FLOPs** without any loss of performance.
- Clear descriptions of **design principles, limitations, and future work.**

Lastly, we would like to remind you about several cases of the technical advances in this field: (1) Transformers → ViT adapted to vision via patchifying images, adding 2D positional priors, and processing pixel-space token sequences; (2) Mamba → VideoMamba extended state-space sequence modeling with spatio-temporal tokenization and long-horizon mixing. Likewise, we present  VideoTitans, motivated by Titans for NLP. VideoTitans adapts memory to video through video-level (history-aware) gradient surprise, patch-level memory commitment, and gated fusion of sliding-window short-term context with long-term memory, along with persistent task priors—achieving linear scaling while preserving temporal coherence. We believe that this work becomes a monument to video prediction task.

---

### Decision · Program_Chairs · 2025-09-17

**Decision:**

Accept (poster)

**Comment:**

## Summary

This paper proposes VideoTitans, an approach for video prediction (predicting future frames conditioned on past frames), the main idea is to adapt the gradient-based memory originally developed for NLP (Titans) into video prediction. Experiments are conducted on various benchmarks to demonstrate the efficiency of the proposed approach in both accuracy metrics (MSE, MAE) and FLOPs.

## Ratings
After rebuttal and discussion, 1 reviewer recommends a rejection and two other reviewers recommend borderline acceptance.

## Reasons to accept the paper
* The idea of adapting Titans architecture from NLP into video prediction is interesting and has potential impact.
* Detailed ablation studies of memory depth, attention types, and integration strategies clarify design choices and trade-offs.
* The FLOPs reduction and computational efficiency are demonstrated across diverse benchmarks.

## Reasons to reject the paper
* The experimental setup for long-range video prediction is unconvincing.
* A limitation is that the paper directly ports the Titans memory module from NLP to video prediction with minimal adaptation. There is no domain-specific modification to account for the spatiotemporal nature of video data.
* Missing comparison with the latest, most advanced video prediction models.


## Discussions
- During author-reviewers, reviewers-AC discussion, the author(s) had addressed most of the concerns including comparing with latest methods, explaining the adopting / changing from NLP Titans.
- AC finds the author(s) response with respect to the long-range video prediction is reasonable as it follows the standard setup used in previous work.
- The author(s) response in explaining the technical contributions in adopting NLP Titans to video prediction is still borderline, there are some domain specific adaptations.

## Decision
AC reads all reviews and joins in the discussions with reviewers. AC believes that the potential and solid performance can outweigh the limitation of novelty. The paper is further discussed with SAC and SAC also agrees to accept the paper. Both SAC and AC recommend to accept the paper.

To the author(s): please incorporate all feedback from reviewers in your final camera-ready version.